# Automated Fluid Intake Detection Using RGB Videos

**DOI:** 10.3390/s22186747

**Published:** 2022-09-07

**Authors:** Rachel Cohen, Geoff Fernie, Atena Roshan Fekr

**Affiliations:** 1KITE Research Institute, Toronto Rehabilitation Hospital, University Health Network; 550 University Ave, Toronto, ON M5G2A2, Canada; 2Institute of Biomedical Engineering, University of Toronto, 164 College St, Toronto, ON M5S 3G9, Canada; 3Department of Surgery, University of Toronto, 149 College Street, Toronto, ON M5T 1P5, Canada

**Keywords:** artificial neural networks, fluid intake monitoring, image recognition, video signal processing, intake gesture detection

## Abstract

Dehydration is a common, serious issue among older adults. It is important to drink fluid to prevent dehydration and the complications that come with it. As many older adults forget to drink regularly, there is a need for an automated approach, tracking intake throughout the day with limited user interaction. The current literature has used vision-based approaches with deep learning models to detect drink events; however, most use static frames (2D networks) in a lab-based setting, only performing eating and drinking. This study proposes a 3D convolutional neural network using video segments to detect drinking events. In this preliminary study, we collected data from 9 participants in a home simulated environment performing daily activities as well as eating and drinking from various containers to create a robust environment and dataset. Using state-of-the-art deep learning models, we trained our CNN using both static images and video segments to compare the results. The 3D model attained higher performance (compared to 2D CNN) with F1 scores of 93.7% and 84.2% using 10-fold and leave-one-subject-out cross-validations, respectively.

## 1. Introduction

The health and safety of older adults who choose to remain living in their homes is a top priority for our society. One major challenge for this population is ensuring they are properly hydrated throughout the day. Dehydration is more common and dangerous in older adults, leading to adverse complications such as hospitalizations and death. Several factors cause older adults to be at risk for dehydration. As one ages, the total water content in the body is naturally reduced, leading to more severe consequences if dehydrated [1,2]. In addition, the ability to feel thirst diminishes as we age, and when this is compounded with cognitive impairments, it may cause older adults to forget to drink [1,3]. Some worry that drinking will lead to frequent urination or urinary incontinence episodes [4].

In seniors, the common dehydration detection methods and gold standards are all highly contested [5,6]. Due to the controversy or invasiveness of the detection methods, healthcare providers strongly suggest encouraging seniors to have regular fluid intake to prevent dehydration [7]. Several methods exist to monitor liquid intake, including sensors placed in containers, wearable devices, vision or environmental approaches, and sensors embedded in surfaces [8].

Previous studies have shown the feasibility of detecting eating and drinking events using vision-based approaches, such as egocentric images (i.e., body-worn cameras) [9,10,11,12] or regular ambient cameras [13,14,15]. However, many studies have focused on classifying individual frames as drinking events. This can lead to a less robust system, as it may be difficult for this model to detect all types of containers in individual frames. With current advances in computer processing and deep learning techniques, it is now possible to classify a series of frames, i.e., a video segment (3D). For example, Bi and Kotz used a head-mounted camera (pointed at the subject’s mouth) to train a 3D convolutional neural network (CNN) for detecting eating events with a 90.9% accuracy and 78.7% F1 score. They compared 2D CNNs, 3D CNNs and SlowFast algorithms, where the latter two significantly outperformed the 2D models [16].

Notably, Roaust and Adam compared 2D and 3D deep learning models to classify intake gestures during a meal [13]. In 102 subjects, they compared models that account for temporal context (3D CNN, 3D LSTM, two-stream, and slow-fast) to the baseline using individual frames (2D models). They also compared using the raw frames to using motion, in the form of frame-to-frame optical flow processing, as input features and found that frames were superior. They achieved an F1 score of 85.8% using a 360-degree camera with the video data. Using the same dataset, by fusing inertial and video data, Heydarian et al. found a similar maximum overall F1 score of 87.3% [14]. However, both [13] and [14] only analyzed subjects during mealtime and no other activities, and the subjects were always in the same location relative to the 360-degree camera. Both studies performed binary classification to classify intake and non-intake, where intake included both food and drink intake.

Similarly, Iosifidis et al. used 3D fuzzy vector quantization and linear discriminant analysis to classify eating and drinking [15]. They also used multiple frames to classify video segments. This model achieved an overall classification accuracy of 93.3%; however, they only examined images from the frontal view and only included data during meal intake from four participants [15].

Conventionally, static frames have been used for image classification; however, with the recent advances and accessibility in computing power, using 3D models to classify activities using multiple frames rather than just a single snapshot in time is now possible. This has resulted in more accurate human action recognition, including drinking events [17]. This paper presents a drinking detection system using an off-the-shelf RGB ambient camera. It compares this task’s accuracy obtained from 2D and 3D CNN algorithms. Although studies have analyzed 3D CNNs for eating gestures, none have specifically focused on drinking intake. Additionally, to our knowledge, no papers have used a robust dataset from a home environment that includes different activities of daily living (ADL) and mealtime gestures.

Section 2 of this paper outlines the data collection protocol and neural network configuration. Section 3 presents the results and discussion to compare the 3D and 2D models for this application as well as future works and limitations. Finally, Section 4 concludes and summarizes the findings.

## 2. Materials and Methods

### 2.1. Data Collection

This paper used a GoPro Hero 9, which captured data at 60fps with a resolution of 2704 × 1520. The study took place in HomeLab, a simulated one-bedroom apartment at KITE Research Institute, shown in Figure 1. The dashed yellow rectangle in this figure represents the area used in this study (living room and dining room), and the red circles represent the cameras’ locations. Ethical approval for the study was obtained from the KITE-TRI-UHN Research Ethics Board (21-5132) on 21 August 2021, and participants gave written informed consent prior to study participation. Five male and four healthy female participants with a mean age of 24 years were recruited to perform two experiments: (1) a controlled drinking scenario (repeated twice) and (2) other activities scenarios, including eating and drinking. The subjects were asked to drink using their dominant hand. The order of the experiments was randomized.

In the controlled drinking scenario, subjects were told which container to drink from and the size of the sip (small, medium and large). The definition of small, medium, and large was subjective and based on the subject’s perception and comfort. The order of the containers, the position of the subject with respect to the camera, and the sip size were all in random combination. In this phase, the subject drank 12 sips from each of the 12 containers listed in Table 1. This experiment was repeated, but the second time the subject was not instructed on the sip size and was instead told to drink a single or double sip. A double sip was defined as a sip where the subject swallowed twice without placing the container down. In the scripted activity scenarios, the subjects were told to perform 12 drinking events (one from each container), 9 eating events (3 of each type in Table 1), and 12 ADLs, as listed in Table 1. They were not given instructions on how to perform the activities, nor were they told the sip size.

These were all in random order and a random combination with a location around the HomeLab’s living and dining room. For the eating activities using a fork, the subject was instructed to bring their lunch that could be consumed with a fork to give a varied dataset. For the eating activity with a spoon, the subjects ate yogurt or cereal, and for the hand-eating activity, the subject consumed crackers, apples or bananas. The subject was instructed to hold a button for the entire duration of their sip to mark the ground truth. The researcher also manually labelled all activities’ start and end times.

### 2.2. Data Processing

The videos from the GoPro camera were first down-sampled to a lower frame rate of 3 or 6 fps. As the GoPro has a wide field of view, the videos and images were cropped to focus on the person in the frame. For this purpose, we first detected the motion by comparing nearby frames, and then the image was cropped in a square of 1000 × 1000 pixels around this motion. All frames were then resized to 224 × 224 pixels to require less computation during training and to be similar to the dataset the models were pre-trained on. The same region of cropping is kept for the entire video sequence. The entire process is shown in Figure 2.

### 2.3. Neural Network Configurations

Our algorithm aims to perform a binary classification identifying drinking and non-drinking events. A 2D CNN architecture was used to classify individual frames, whereas to classify videos (windows of multiple frames), a 3D CNN was proposed. We used Transfer Learning to build both 2D and 3D models. Transfer learning is a technique in which an existing model trained on datasets of a similar application is retrained with our dataset. This technique reduces the amount of data required and, in many cases, increases the accuracy compared to models built from scratch, as the original models were trained on large amounts of data. In this paper, we tested 8 state-of-the-art pre-trained models for our 2D and one for our 3D CNN. To determine which parameters yielded the highest performance, various models were trained and tested by altering parameter combinations as follows:Window size for 3D: We extracted inputs of specific window sizes for the video data. We tested two different window sizes, 3 and 10 s.Frame rate for 3D: The frame rates were chosen as 6fps and 3fps to compromise computational cost and accuracy.Imbalanced data: In all cases, the data was imbalanced, as there were more non-drinking events. We compared two common methods to handle the imbalance: (1) class weights and (2) random under-sampling. Class weights involve applying a larger weight to the minority class (drinking class) so that it places more importance if these are misclassified during the training process. Under-sampling involves decreasing the majority class to be the same size as the minority class by randomly choosing inputs and discarding the rest. Oversampling, which involves randomly sampling and duplicating from the minority class until the classes are balanced, was also attempted.Layers trained: Either all the layers or only the top layer (known as feature extraction) were trained. After this, fine tuning was performed, which involves freezing the weights of all layers except the top layers (2/3rds) to refine the prediction.Validation method: A 10-fold cross-validation and leave-one-subject-out (LOSO) were both tested and compared.Pre-trained models: For the 2D models, 8 state-of-the-art pre-trained models were tested for the image data to determine the best one. These include DenseNet169, DenseNet121, InceptionResNetV2, InceptionV3, Xception, MobileNetV2, NasNetLarge, ResNet. Only one state-of-the-art model was considered for the video data, proposed by Carreira et al., called Inflated 3D Conv Nets (I3D), shown in Figure 3 [18]. This model is based on Inception-V1 but with inflated layers to add a temporal dimension. This was originally used with 71.1% accuracy to classify 400 human activities and outperformed other temporal classifiers on common benchmark datasets. This is a very deep, spatiotemporal classifier. This model was effective in other papers on video data [19,20,21]. This paper used the RGB-I3D network, which is pre-trained on the ImageNet dataset (https://www.image-net.org/ (accessed on 30 August 2022)) and the Kinetics dataset (https://paperswithcode.com/dataset/kinetics (accessed on 30 August 2022)). We added a dropout layer and an early stopping mechanism to prevent overfitting.

## 3. Results

Due to the imbalanced nature of our dataset, the main classification metric considered is the F1 score, which is the harmonic mean of the precision and recall, placing equal weight on both metrics. Based on the combination of parameters listed in Section 2.3, we have trained 48 3D models (2 window sizes, 2 frame rates, 3 sampling methods, 2 options of numbers of layers trained, and 2 validation types) and 96 2D models (3 sampling methods, 2 options of the number of layers trained, 2 validation types, and 8 models). This paper presents the results of the combination that yielded the highest F1 score (best/top models).

Overall, Xception, DenseNet169 and DenseNet121 had superior results for the 2D models. For the 3D models, using 3fps and 6fps gave similar results, and thus, 3fps was chosen as it requires less computational power and time. The 3 s windows yielded higher F1 scores for the 10-fold validation, but the 10 s window was superior for the LOSO validation. No overall trend was observed comparing feature extraction versus training all the layers. Figure 4 shows a distribution of the F1 score for all models trained. For LOSO, in most cases (blue in Figure 4), 3D is superior. For the 10-fold validation (orange in Figure 4), the best models are very similar, but the range of F1 scores is larger for the 2D models.

The confusion matrices for the best/top 3D and 2D models are shown in Figure 5, i.e., the model with the highest F1 score for that validation method. Table 2 shows the performance metrics and parameter descriptions of the best 3D and 2D models.

Figure 6 shows the top models’ receiver operating characteristic (ROC) curves. The ROC graph’s area under the curve (AUC) measures the ability to predict the classes properly. The “chance” line represents an AUC = 0.5, meaning the classification is equal to the performance of random guessing. Figure 7 shows the precision-recall curve for the best models, which plots precision against recall for different probability thresholds. These are more appropriate for imbalanced datasets, such as the “class weights” option. The 3D models have a higher AUC in all cases for both ROC and precision-recall curves, showing a superior performance at classifying drink events across the folds and subjects.

To ensure the models were looking into the right location of the images and videos, we deployed a gradient class activation map (GradCAM), shown in two separate examples in Figure 8. GradCAM uses the gradients flowing into the final convolutional layer and creates a heat map of the important regions of the image [22]. The test data, either image or video, are passed into the model with the final convolutional layer. The gradients with respect to the model loss are calculated to determine the regions on the frames which contribute most to the prediction.

## 4. Discussion

As expected, the 10-fold cross-validation performs better than the LOSO validation. This is even more pronounced with the 2D models. Notably, to compare the 2D and the 3D algorithms, the exact frames present in the 3D data were extracted as single-frame inputs for the 2D CNN models. This led to some images being very similar to each other in the dataset. Therefore, comparing the LOSO results of the 2D and 3D models is more useful, as there is no chance of data leakage.

Overall, the 3D video classification performed better than the 2D, though this difference is small when analyzing 10-fold cross-validation. The 3D top model obtained an F1 score of 94.23%, while the 2D top model obtained an F1 score of 93.7% (Table 2). When analyzing LOSO, the difference is larger, from 84.2% for the 3D model to 73.6% for the 2D model (Table 2). It is expected that the 3D models would be superior, as they detect the motion to drink rather than an individual image with no motion information. The dataset included a wide variety of drinking vessels, including those with straws that were not easily visible to the camera. The 3D data takes the entire motion as an input, so frames approaching the mouth but not yet a drinking event are not misclassified, as they are all within the same input event and classified together. Additionally, for the straw vessel, the motion of the cup to the mouth is still present, so the 3D algorithm can detect it while the straw itself is not present and difficult for the 2D model to detect.

For the top LOSO models, the 2D models have higher true negative rates, meaning it is good at predicting non-drink events, but the 3D models have a higher true positive rate. However, the 3D models have lower false negative rates in nearly all cases but often have higher false positive rates, meaning they may predict a drinking event that did not happen.

As seen in Figure 7 in LOSO validation, subject 2 had a lower performance than the other subjects for the 2D models, which is particularly evident in the precision-recall curves. It is worth noting that this subject was the only left-handed drinker, which could be a factor for higher misclassification. Therefore, adding more left-handed drinkers to the training dataset may improve the model’s performance. Additionally, compared to the other subjects, this subject often sat in such a way that their face was slightly occluded from the camera, and their long hair covered more of their face, as seen in Figure 9. Although many subjects sat in this orientation, due to the randomization, this subject sat in this position more often, and therefore their long hair, left-handedness and more diagonal positioning are factors that likely reduced the model’s ability to predict the drinking event. This finding is similar to Costa et al., who also found detecting drinking in the left-hand challenging [23]. When excluding this subject, the F1 score increases by 5.4% for the best 2D model. However, the 3D models were able to correctly classify this subject’s drink events in the LOSO validation, highlighting the robustness of this method.

Another observation was that, in 2D data, the events most often misclassified are the movements leading up to a drink. This includes, for example, the few images before and after a drinking event when the person is lifting the container, but the container is not yet touching the mouth. These were labelled as “null” events, as only the frames with the container touching the mouth were considered a drinking event. For the top 2D 10-fold model, these errors accounted for 93% of the misclassified events (>5 folds misclassified the image); however, this trend was not observed for the LOSO models. When these images were removed from the dataset and the models were retrained, the LOSO F1 score increased by 18% to 81%. However, in real scenarios, people may lift the drinking container to their mouths without drinking, for example, to blow on a hot drink, which is not practical. Again, the 3D model proves to be more robust for this scenario.

As observed in the class activation maps in Figure 8, the model looked at the person and the vessel when the image or video was properly classified. In contrast, in the misclassified events, we observed several instances where the model focused on objects in the background, such as the artwork on the wall. An example is shown in Figure 10, where the participant is near the edge of the cropped frame. This shows that the environment does play a key role in the model’s performance. In the future, we will add an object detection/tracking algorithm to ensure only the objects of interest are in the input data.

As seen in Figure 5, there is a higher rate of false positives than false negatives, especially in the LOSO validation. This means the system could overestimate the total amount drank during the day. This could be a concern, as the person would not be reminded to drink when needed. Therefore, in this specific application, selecting a final model with a lower number of false positives may be beneficial even with larger false negatives, which may result in a lower F1 score.

The downside of using 3D models is the high computational cost of training the models. A larger amount of memory is needed for the same batch size when training, as more frames are input for each iteration. For real-time implementation, the 3D model would need to store frames for the initialized window length (in this case, 3 s) before starting the classification. The test computational time for the 3D models was 1.3 s compared to the test time for the 2D model, which was 0.4 s. In a real-world deployment, both of these test times are practical, even for the 3D model with a built-in lag while collecting the input frames.

A limitation of this study was that some hyperparameter combinations could not be tested due to memory constraints, such as larger batch sizes with oversampling. However, multiple batch sizes were still evaluated to mitigate this issue, and oversampling was attempted where possible (i.e., with smaller batch sizes). As the oversampling provided a similar result to the class weights or under-sampling, we excluded it from the results. A smaller batch size requires less memory to train the model but increases the training time and usually improves the generalizability of the algorithm.

Other well-known limitations of using vision-based approaches for drink classification are that the person must be in the environment, visible to the camera, and should not be occluded. There is also a privacy concern, as many do not wish to have cameras placed in the home. To meet this concern, in the future, we plan to investigate a depth camera as a more privacy-pervasive solution compared to regular RGB cameras. Another limitation is that the algorithm only detects when the container is brought to the mouth, but it does not know if the liquid was actually swallowed. More investigation needs to be done to address this challenge in a future study.

As seen in Table 3, this work builds upon the previous literature by providing a 3D CNN model to detect drinking events in a simulated home environment with a dataset including multiple orientations, actions, and backgrounds. It obtains comparable or superior results to the previous works. The proposed model showed the ability to reliably detect drinking events in a simulated home environment, with unscripted activity scenarios that included different activities in multiple orientations. This makes the model robust enough to be applied in seniors’ homes to monitor their hydration levels.

Future work can collect data from more participants and include a free-living scenario to expand the dataset further. Finally, future work could include a classification for eating, particularly with a spoon, as it can contribute to the subject’s hydration. Additionally, a system should be created that prompts the user to drink based on the number of sips detected by the model.

## 5. Conclusions

This paper uses an RGB camera to demonstrate and compare different deep learning models to detect drinking events in a physically simulated home environment. The models were trained with both individual frames (2D CNN) and a set of frames (3D CNN) for the binary classification of a drinking event. It was found that using multiple frames as the input (3D CNN) can improve the F1 score compared to using individual frames, particularly in LOSO cross-validation. Our preliminary data collection obtained an overall F1 score of about 84% with the 3D network. The 3D CNN models are more robust against different drinking containers, orientations, and locations, as they detect the motion of the hand movement during the whole event.

## Figures and Tables

**Figure 1 sensors-22-06747-f001:**
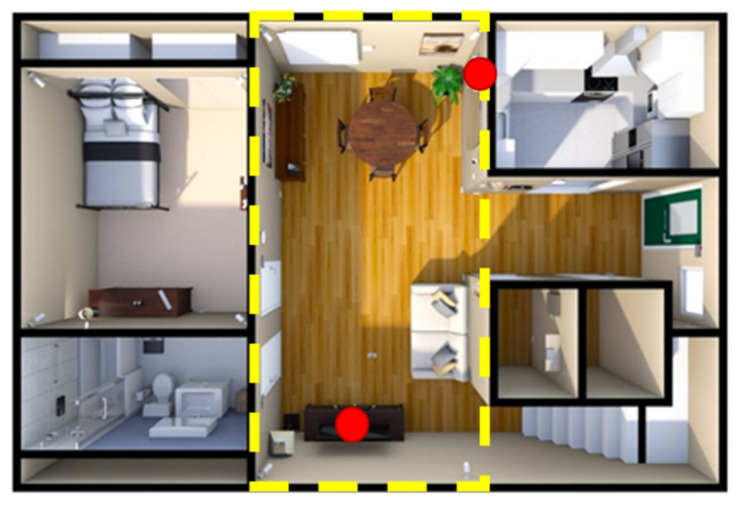
HomeLab layout at KITE Research Institute, where the yellow box represents the areas in the field of view of the cameras, and the red represents the location of the cameras.

**Figure 2 sensors-22-06747-f002:**
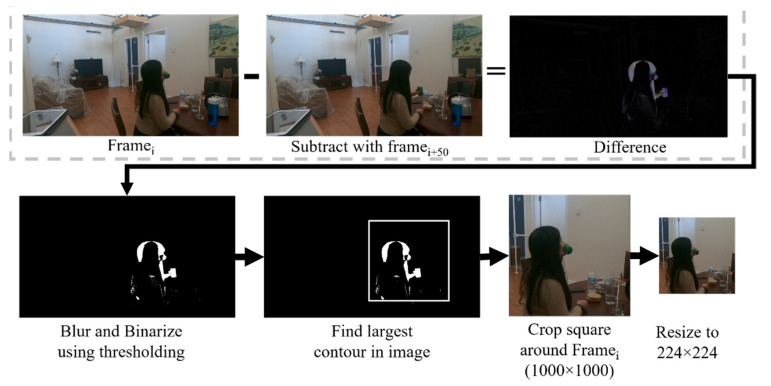
Description of the data processing method used in this paper.

**Figure 3 sensors-22-06747-f003:**
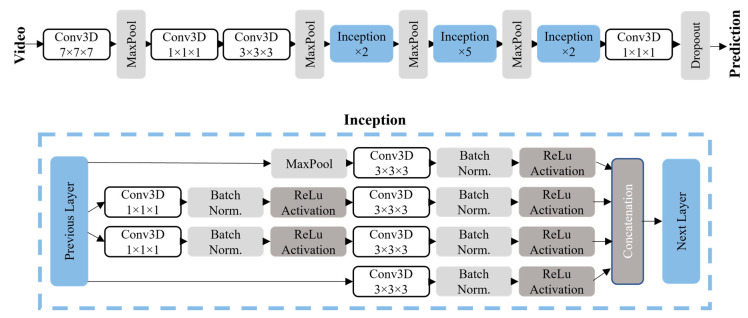
Model architecture for the I3D model, proposed by Carreira et al. The convolutional layers are all followed by batch normalization and ReLu activation. The final prediction uses SoftMax activation.

**Figure 4 sensors-22-06747-f004:**
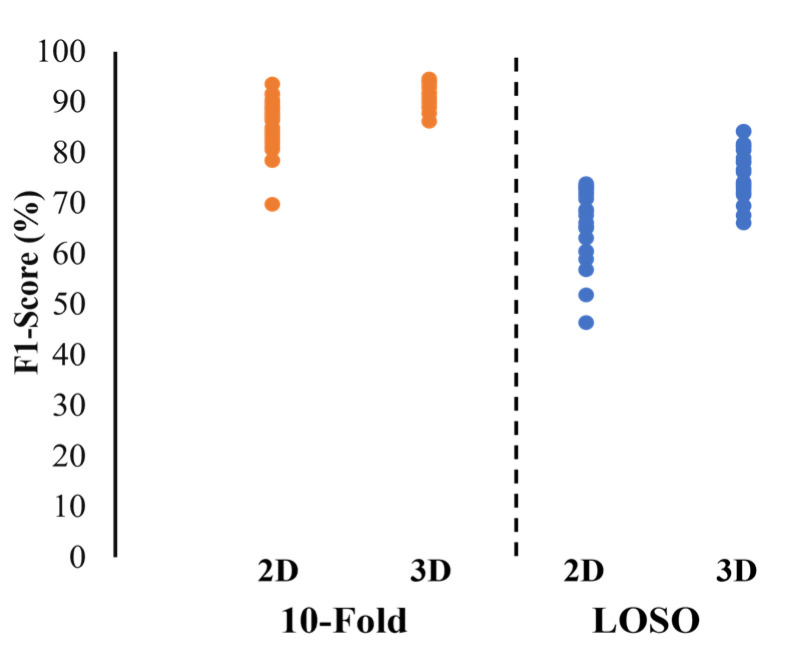
Distribution of F1 score for all models trained with various parameters.

**Figure 5 sensors-22-06747-f005:**
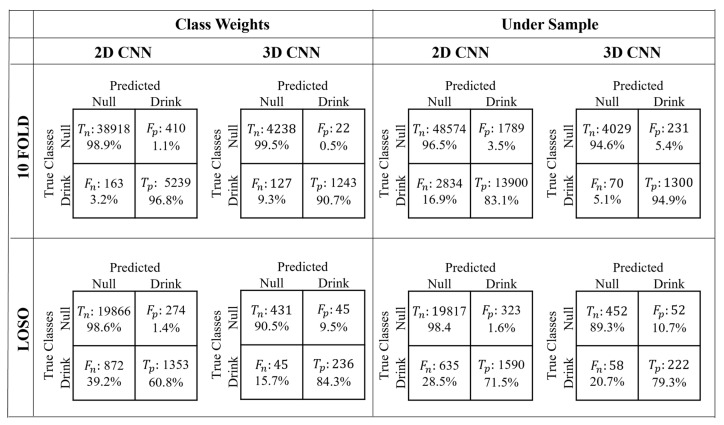
Confusion matrices from the 10-fold and LOSO validation, trained by using class weights and under-sampling for the 3D and 2D classification models. T_n_, F_p_, F_n_, and T_p_ refer to true negative, false positive, false negative and true positive.

**Figure 6 sensors-22-06747-f006:**
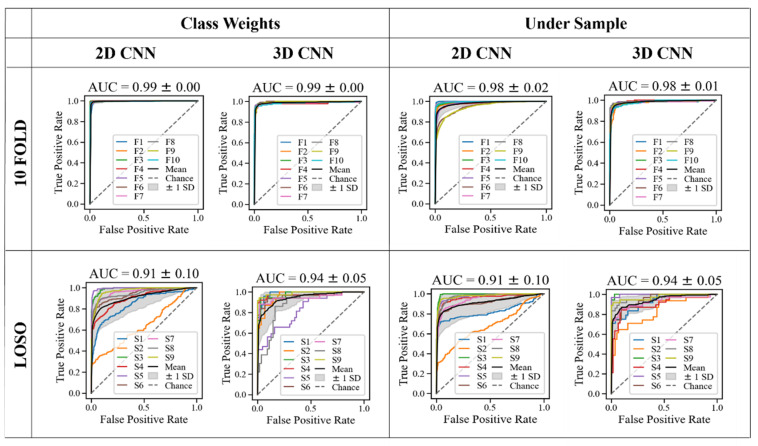
Receiver operating characteristic (ROC) curve for the top models. F_x_ refers to the fold number x and S_x_ refers to the subject number x who has held out in the LOSO validation.

**Figure 7 sensors-22-06747-f007:**
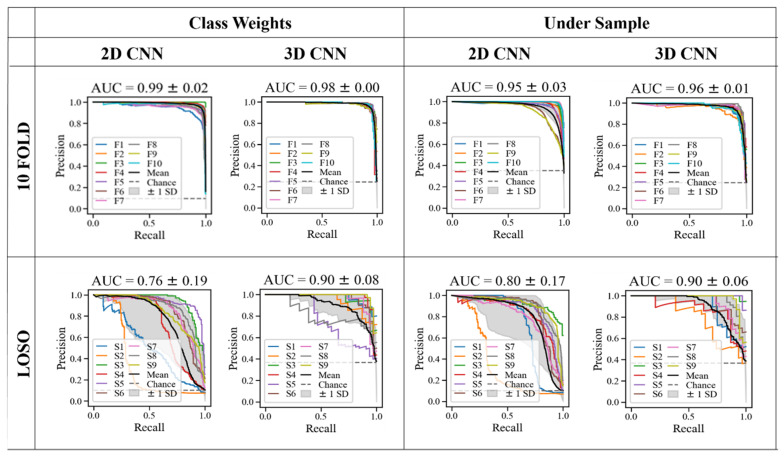
Precision-recall curve for the top models.

**Figure 8 sensors-22-06747-f008:**
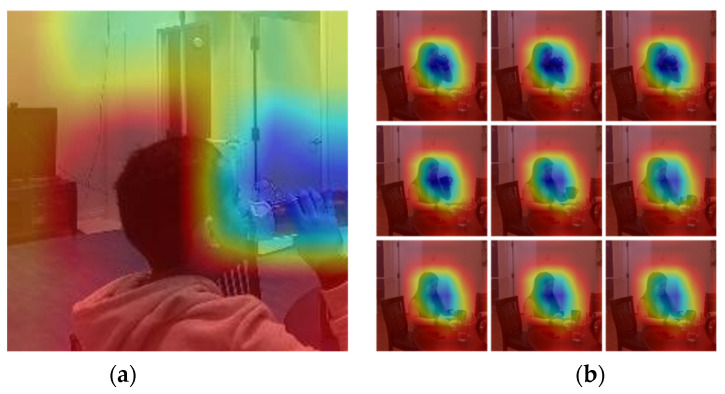
Gradient class activation maps for (**a**) 2D example and (**b**) the frames of a 3D example.

**Figure 9 sensors-22-06747-f009:**
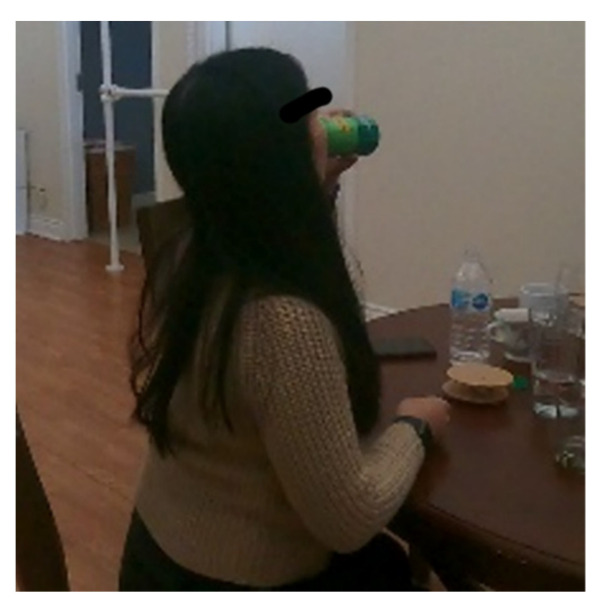
Subject 2 sitting in a position where their face is slightly occluded from the camera.

**Figure 10 sensors-22-06747-f010:**
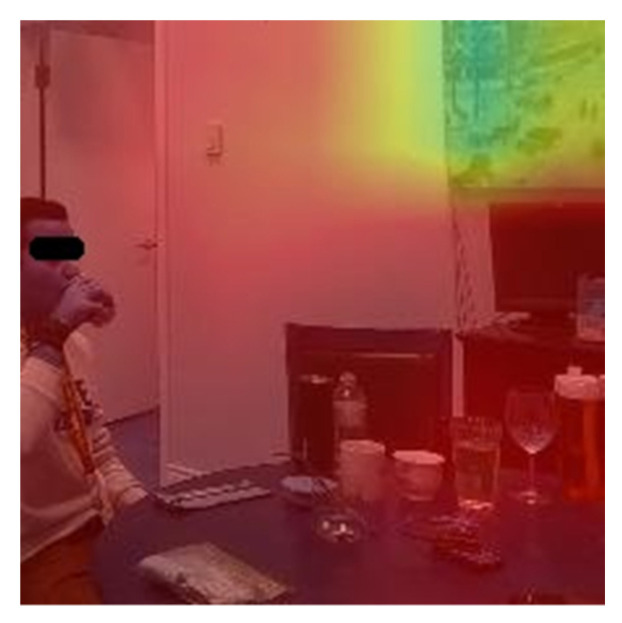
Example of a misclassified frame where the model is detecting the background environment.

**Table 1 sensors-22-06747-t001:** List of all activities performed in our study.

Drinking Container	ADLs Activities
A teacup with hot liquid A coffee mug with hot liquid Two metal commercial water bottle A plastic, disposable water bottle A can A glass tumbler with ice water A glass tumbler with colored liquid (pop or juice) A plastic, short-colored cup with water A wine glass with colored liquid (pop or juice) Two glass tumblers with a straw	Scratching their head and face Pointing the TV remote and watching TV Doing their hair/touching their head Using a laptop Using a smartphone (calling and texting) Pouring water from a kettle Stretching Washing the counters Putting on and taking off a jacket Walking around the apartment Talking to the researcher Writing Folding laundry Eating with a fork, spoon and hands 3x each

**Table 2 sensors-22-06747-t002:** Performance metrics for the best models. The values in the brackets represent the standard deviations for each metric across the folds.

Model	Validation	Model Description	Sampling	Accuracy (%)	Precision (%)	Recall (%)	F1 Score (%)
3D	10-FOLD	Window size: 3s, Sampling rate: 3fps, Batch size: 32, with feature extraction	Class weights	97.1 (0.6)	98.3 (1.2)	90.7 (2.3)	94.3 (1.2)
			Under-sample	94.6 (2.6)	85.8 (8.0)	94.8 (1.3)	89.8 (4.3)
	LOSO	Window size: 10s, Sampling rate: 3fps, Batch size: 16	Class weights	88.6 (9.4)	88.6 (12.5)	85.8 (20.7)	84.2 (14.8)
			Under-sample	86.3 (7.7)	86.5 (12.9)	80.3 (18.1)	80.8 (11.1)
2D	10-FOLD	DenseNet121, Batch size: 32	Class weights	98.7 (0.64)	91.4 (5.4)	96.2 (3.5)	93.7 (3.3)
			Under-sample	93.2 (4.4)	81.4 (16.2)	74 (24.4)	75.4 (20.6)
	LOSO	Xception, Batch size: 16, with feature extraction	Class weights	95.0 (1.9)	86.3 (14.8)	60.7 (19.7)	68.2 (17.0)
			Under-sample	95.7 (1.03)	83.2 (6.4)	70.0 (20.7)	73.6 (14.3)

**Table 3 sensors-22-06747-t003:** Comparison with the literature.

Ref.	Videos (3D)	Accuracy (%)	F1 Score (%)	Cross Validation	#Subjects	Camera Direction/Actions
Iosifidis [15]	Yes	93%	-	LOSO	4	Frontal/mealtime only
Rouast [13]	Yes	-	85.8%	Holdout	102	360-degree camera/mealtime only
Proposed	Yes	97.1% 88.6%	93.7% 84.2%	10-Fold LOSO	9	Multiple orientations in simulated home/eating, drinking, and ADLs

## Data Availability

The data are not publicly available due to restrictions in the ethical agreement on data sharing.

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
