# Peer review of "Automated Fluid Intake Detection Using RGB Videos"

_sensors, 2022, doi:10.3390/s22186747_

Round 1

Reviewer 1 Report

Yes, the dehydration is a common, serious issue among older adults so it is important to drink fluid regularly to prevent dehydration and the complications that come with it. I found this paper interesting because it is connected with our daily life. Few suggestions will be here to improve this paper at some extent:

(1)    How your approach is better over vision-based approaches? Justify.

(2)    How the results vary when your approach (i.e. 3D CNN based video segment) is used in place of static frame approach? Explain and incorporate at suitable place in the text.

(3)    Add one paragraph about the organization of this manuscript in Introduction Section.

(4)    Incorporate one Section before Conclusion Section about the significance of your contribution in few lines clearly.

(5)    It will be helpful for the researcher in same area, if you will add few line about future directions.

Reviewer 2 Report

This paper demonstrates and compares different deep learning models to detect drinking events in a home environment using an RGB camera. In consequence, the authors clarified that the 3D CNN models are more robust against different drinking containers, orientations, and locations, as they detect the motion of the hand movement during the whole event. The purpose of the research is meaningful and the paper is well written. However, improvements are needed for the reader of this paper.

Some minor issues are listed below.

1.    Line number 10 of the abstract, “track” should be better replaced with “tracking”.

2.    In order to clarify the significance of this paper, please specify the characteristics of this research compared to the conventional method at the end of the introduction.

3.    Please clarify the structure of the CNN used in your research. For example, the number of convolutional layers and pooling layers, the activation function, the number of units in fully connected layers, etc.

Reviewer 3 Report

The paper is attempting to find solutions for a rather complicated problem, when tackled based on camera data. The results are very preliminary and lots of situations that could not be handled have been eliminated, although the test environment is already coming closer to real life conditions than in previous studies that are however more careful in their evaluation procedures. The paper deserves to be published, but with major modifications and a very clear statement that the work presents preliminary results, both in the abstract and in the conclusions.  There are not enough quantitative descriptive data on the methods used.  The quantitative performance percentages in the text should be systematically checked on their compatibility with the numbers in Figures and tables.

More detailed comments below:

Abstract

The last sentence is not clear. I propose the following modification: The 3D model attained higher performance (compared to the 2D CNN) with F1-scores of 93,7 % and 84.2 % using respectively 10-fold and Leave One Subject Out cross validations.  # The percentages are now compatible with Table 3 #

Section 1:

presented -- > present

computing processing-- > computer processing

Unclear formulation: optical flow between the frame to frame before it – > frame-to-frame optical flow preprocessing

Both performed – Both studies performed

livings-- > living

Section 2

preform -- > perform

found in Table 1 - > listed in Table 1

**** Section 2.2 mentions two frame rates, but they do not appear anymore in Section 3 on the results

**** A figure, illustrating with an example, on how you do the image manipulation to arrive at 224x224 images should be added.

**** Section 2 .2 mentions 8 pretrained state-of-the art models for the 2D and 1 for the 3D case. But there is no reference to these 8 models. Please name them and refer to them and give some quantitative results not only for the best model but also for the others. For the 2D case, the reader has no clue on which network was used.

**** More quantitative data are needed about the refinement training

naturally spatiotemporal classifier -- > spatiotemporal classifier   # What do you mean by naturally? #

ImageNet -- > ImageNet (https://www.image-net.org/ )

Kinetics -- > Kinetics data set (https://paperswithcode.com/dataset/kinetics)

Section 3

**** What do you mean by 72 3D models and 96 2D models. Where do these numbers come from?

Figure 2:  Although clear, for the sake of completeness, add the meaning of Tn, Fn, Tp and Fp in the caption

Top models? What do you exactly mean by top models?  Clarify

**** Figures 3 and 4: SD seems intuitively very small compared to the spread of the curves. Clarify this. S1- S9: add in the caption that these identify the subject that was left out In LOSO. F1—F10: add in caption that FX indicates fold number X

exaggerated -- > pronounced

**** Where somewhere do I find the numbers 95,2%, 93,7 % 84,2 % and 63,3 %? Refer to a Figure or a Table?

their -- > his/her   # 4 times and in the caption of Figure 5, since you are talking about one Subject 2 #

This is similar -- > This finding is

**** How have you designed the activation map? More info needed.

the vessel together -- > the vessel together (see Figure 6)

“A larger amount …classification” When you speak about implementation in this phrase, are you still referring to training or not? Be clear on that.

The test time for… do you mean the test computational time?

**** The whole paragraph “A limitation … results” is unclear. “Some combinations could not be tested”, combination of what?  Oversampling of what? “Multiple batch sizes were still evaluated”, how does batch size alleviate some problems. Can you be specific, by giving quantitative examples or data on the batch sizes.

A robust data set is somewhat strange as a terminology. I would drop “robust” and “robustness of the”

Oversampling was attempted where possible. When was it possible?

Section 5 (there is no Section 4)

home -- > physically simulated home

When you add quantitative results,  as suggested in my comments, it would be adequate to consider splitting  “Results” and “Discussion” in two sections

Round 2

Reviewer 3 Report

Congratulatrions with the pertinent replies to my questions and the very good formulations of the text improvements. You succeeded in giving a lot of usfeul informations in a rather short paper.

While verifying your corrections I came across some very minor things that you might correct:

Line 193: as requires -- > as it requires

Line 237: As expected, the 10-Fold cross validation performs better than the LOSO validation. - > As expected, the 10-Fold cross validation produces higher performance values than the LOSO validation.

Line 308: for 2D model -- > for the 2D model

Line 326: correct As seen in Error! Reference source not found.3, this work builds upon